# Estrogens in Adipose Tissue Physiology and Obesity-Related Dysfunction

**DOI:** 10.3390/biomedicines11030690

**Published:** 2023-02-24

**Authors:** Alina Kuryłowicz

**Affiliations:** 1Department of Human Epigenetics, Mossakowski Medical Research Centre PAS, 02-106 Warsaw, Poland; akurylowicz@imdik.pan.pl; Tel.: +48-226086591; Fax: +48-226086410; 2Department of General Medicine and Geriatric Cardiology, Medical Centre of Postgraduate Education, 00-401 Warsaw, Poland

**Keywords:** estrogen, estrogen receptor, adipose tissue, adipokines, obesity

## Abstract

Menopause-related decline in estrogen levels is accompanied by a change in adipose tissue distribution from a gynoid to an android and an increased prevalence of obesity in women. These unfavorable phenomena can be partially restored by hormone replacement therapy, suggesting a significant role for estrogen in the regulation of adipocytes’ function. Indeed, preclinical studies proved the involvement of these hormones in adipose tissue development, metabolism, and inflammatory activity. However, the relationship between estrogen and obesity is bidirectional. On the one hand-their deficiency leads to excessive fat accumulation and impairs adipocyte function, on the other-adipose tissue of obese individuals is characterized by altered expression of estrogen receptors and key enzymes involved in their synthesis. This narrative review aims to summarize the role of estrogen in adipose tissue development, physiology, and in obesity-related dysfunction. Firstly, the estrogen classification, synthesis, and modes of action are presented. Next, their role in regulating adipogenesis and adipose tissue activity in health and the course of obesity is described. Finally, the potential therapeutic applications of estrogen and its derivates in obesity treatment are discussed.

## 1. Introduction

Recent years have changed our understanding of the role of adipose tissue in human health and disease. Currently, it is not considered only as energy storage but an active endocrine organ that may modulate the function of other tissues and systems [1]. Obesity-related excessive accumulation of lipids causes changes in the metabolism of adipocytes, leading, among other things, to the dysfunction of the mitochondria and the associated endoplasmic reticulum stress [2]. As a result, the expression of several genes in the adipocyte changes unfavorably, and thus the profile of substances secreted by adipose tissue (adipokines). This process, known as adipose tissue dysfunction, is believed to underlie the development of insulin resistance predisposing to glucose intolerance and several other obesity-related chronic complications, affecting virtually all organs and significantly deteriorating the quality of life, which constitutes a serious social and economic problem [3].

Importantly, the activity of adipose tissue depends on its type (white, brown, beige) and depot (visceral, subcutaneous, perivascular, etc.). Therefore, the health risk of an obese individual is determined not only by the total amount of adipose tissue but also by its distribution and metabolic activity [4]. White adipose tissue (WAT) distribution varies by gender. Men tend towards visceral (android) obesity, which is associated with increased insulin resistance and cardio-metabolic risk. Overall, women have higher adiposity than men; however, their adipose tissue accumulates favorably in the subcutaneous depot, which is associated with a lower risk of obesity-related complications [5]. This finding points to the role of estrogen in regulating adipose tissue distribution [6,7]. Moreover, recent years have brought evidence confirming the role of estrogen in regulating adipocytes’ insulin sensitivity, metabolism, and secretory activity [8,9,10]. Subsequently, estrogen deprivation was linked to an increased risk of obesity and metabolic complications, which can be partially reversed by hormone replacement therapy [11,12,13,14,15].

This narrative review aims to summarize the role of estrogen in adipose tissue development, physiology, and in obesity-related dysfunction. Firstly, the estrogen classification, synthesis, and modes of action are presented. Next, their role in the regulation of adipogenesis and adipose tissue activity in health and the course of obesity is described. Finally, the potential therapeutic applications of estrogen and its derivates in the treatment of obesity are discussed.

## 2. Estrogens, Their Synthesis, and Mechanisms of Action

### 2.1. Estrogens

Estrogens constitute a group of lipid-soluble steroid hormones that, based on their origin, can be divided into two major categories: endogenous and exogenous. Endogenous estrogens are synthesized in cells of living organisms-both animal and plant (the last are known as phytoestrogens), while exogenous estrogens include pharmaceutical estrogens and xenoestrogens [16].

In humans, four estrogens have been identified: estrone (E1), estradiol (E2, which has two isomers: 17α-E2 and 17β-E2), estriol (E3), and estetrol (E4) [17]. Each form of estrogen presents a different product delivered from cholesterol by a series of reactions (described in the subsequent section). Estradiol (E2) is the predominant estrogen in women during reproduction (both in terms of absolute serum levels as well as in terms of estrogenic activity). After menopause, it is replaced by estrone (E1) synthesized in adipose tissue from adrenal dehydroepiandrosterone. Estriol (E3), formed from E1 through 16α-hydroxylation, is the predominant circulating estrogen during pregnancy. In turn, E4 is an estrogen produced by the fetal liver and therefore, detectable only during pregnancy. Subsequently, E3 and E4 levels are negligible in mature men and non-pregnant women [17,18]. In general, the activity of E2 is about 10-fold higher than E1 and about 100-fold higher than E3 and E4l. Thus, estrogen deactivation can include both conversion from estradiol to less-active forms, such as E1 or E3, and sulfation by estrogen sulfotransferase to the forms which are no longer interacting with estrogen receptors. Therefore, the ratio of circulating estrogens indicates the balance between their synthesis and deactivation [19].

Examples of pharmaceutical estrogens are ethinyl estradiol (EE), a derivative of E2 used in contraceptives and hormone replacement therapy (HRT), and conjugated estrogens (CE) used in HRT. In turn, Bisphenol A (BPA), a synthetic chemical used, among others, in the production of polycarbonate bottles and coatings in cans, is a representative of xenoestrogens [16].

In addition, there is a large group of chemical compounds known as selective estrogen receptor modulators (SERMs) able to bind and interact with estrogen receptors and possess estrogen agonist or antagonist properties at different target tissues, including adipose tissue (described in the following sections) [20]. 

### 2.2. Estrogens Synthesis

Classical steroidogenic tissues, including gonads, adrenals, and placenta, can synthesize steroid hormones de novo from cholesterol, while steroid synthesis in other tissues mainly relies on the conversion of various precursors obtained from circulation [21]. Although the role of adipose tissue as a major steroid conversion site is well established [22], it was also found to be able to initiate steroidogenesis de novo [23,24]. 

The key enzyme for estrogen synthesis is aromatase (CYP19A1) whose expression and activity have been reported in many human tissues including, e.g., endometrium, brain, bone, skin, and adipose tissue [25,26]. The effect of aromatase activity depends on the local availability of its substrates-androgens (Figure 1). For instance, in the ovary, where the main available androgen is testosterone, CYP19A1 activity results in the synthesis of estradiol. In contrast, in the adipose tissue, the main aromatase’ substrate is androstenedione (delivered from the dehydroepiandrosterone (DHEA) and its sulfate both synthesized in adrenals), whose aromatization leads to the synthesis of estrone [24]. The conversion rate of androstenedione into estrone increases with age and adipose tissue volume, and it is higher in women with gynoid than in those with android obesity [27]. Estrone can also be synthesized in the adipose tissue by the oxidation of estradiol in the reaction catalyzed by 17β-hydroxysteroid dehydrogenases (17β-HSD) types 1, 7, and 12. Subsequently, estrone can be converted by steroid sulfotransferase (STS) to estrone sulfate, which is the most significant component of the pool of circulating estrogen. Another enzyme essential for the local availability of sex steroids is hormone-sensitive lipase (LIPE) which hydrolyses fatty acyl esters (FAE) of DHEA and E2 [24,28]. Fatty acyl esterified E2 is its storage form, unable to exert its biological functions. Therefore, E2 esterification/hydrolysis balance is an important regulatory mechanism of biologically active steroid levels [29,30]. Estrogens are eliminated from the body mainly as sulfated and glucuronidated derivatives. The first step in this process involves the generation of hydroxylated derivatives. The hydroxyl group can then be sulfated, glucuronidated, or methylated [31]. 

The most abundant steroids in adipose tissue, regardless of the depot and gender, are DHEA and androstenedione [32,33]. In both women and men, estrogen levels in adipose tissue are lower than their precursors; however, there is a positive adipose tissue/plasma gradient for E1 and E2 [27,32,33,34]. In women, concentrations of E2 in subcutaneous adipose tissue (SAT) are higher than in men, which correlates with a higher expression of estrogen-converting genes [29,30]. Importantly, steroidogenesis in adipose tissue may depend on menopausal and nutritional status. In both, pre-and postmenopausal women, visceral adipose tissue (VAT) is characterized by a higher concentration of E1 compared to the SAT [27,34]. However, while in postmenopausal women obesity is associated with the increased concentrations of E2 in VAT, in premenopausal women it is associated with a higher CYP19A1 activity and subsequent higher estradiol synthesis in SAT [27,34].

### 2.3. Estrogen Mechanisms of Action

Estrogen exerts its biological functions via interaction with its receptors (ERs) which can be both: nuclear and membrane-associated. Nuclear ERs exist in two main forms, α and β, which differ in their tissue expression and function [35]. While ERα plays a stronger physiologic role in females, the activity of ERβ is similar in men and women [36]. Upon ligand binding, ERs undergo conformational changes that allow for the formation of heterodimers and interaction with estrogen response elements (ERE) in the promoter of a target gene [37]. However, ERs can also act in an ERE-independent manner by modulation of co-regulatory proteins and transcription factors that are bound to their cognate responsive elements on DNA. Obesity is associated with a significant decrease in the expression of both nuclear ER subtypes in adipose tissue, while weight loss leads to an increase in ERα and ERβ mRNA levels [38,39]. Importantly, since ER subtypes have a diverse impact on gene transcription, the local ERα/ERβ ratio is critical for the final effect of estrogen action in a particular tissue [9]. The proportion between ERα and ERβ in adipose tissue can physiologically evolve with aging, but also be disturbed by pathological conditions [40]. This is the case with obesity: the adipose tissue of obese individuals of both sexes is characterized by a higher ERα/ERβ ratio compared to the tissues obtained from the normal weight subjects [39,40].

Additionally, estrogen can act rapidly (in a non-classical or non-genomic mechanism) via membrane-associated receptors interacting with other signaling molecules, e.g., G proteins, growth factor receptors, tyrosine kinases (Src), etc. The most prominent form of membrane ER is GPER1 (G protein-coupled membrane receptor1) [41,42].

In turn, in the mitochondria, ERs can modulate gene expression either by direct interaction with the mtDNA or by increasing the activity of manganese superoxide dismutase. In addition, the activity of mitochondria can be modulated by the nuclear ERs that regulate the expression of genes crucial for mitochondrial actions [41,43].

In the context of adipose tissue biology, ERα has been the most extensively studied, while the role of ERβ and membrane-associated forms of ER is much less characterized.

## 3. Estrogens in the Regulation of Adipocyte Proliferation and Fate

Three distinct types of adipocytes have been identified in humans: white, brown, and beige, differing in origin, morphology, and metabolic profile [1]. In adults, the most abundant is white adipose tissue (WAT), composed of white adipocytes, the metabolic and secretion activity of which may differ depending on its location-visceral (VAT) or subcutaneous (SAT) [44,45].

Sexual dimorphism of adipose tissue distribution in humans appears in puberty which indicates the role of sex hormones in its development. Indeed, estrogens have been involved in the regulation of key steps of preadipocyte differentiation, proliferation, and white and brown adipogenesis [7].

### 3.1. Estrogens in the Regulation of Stem Cell Differentiation and Preadipocyte Proliferation

Excess adiposity is primarily considered a result of adipocytes’ hypertrophy. However, studies either with animals on a high-fat diet (HFD) or with ERα mutants pointed to adipocytes’ hyperplasia also as an important mechanism of adipose tissue expansion. Adipose tissue contains stem cells able to differentiate into various mesenchymal cell lineages including bone, cartilage, tendon, and fat, as well as muscle and endothelial cells [46].

In vitro studies suggest that the influence of estrogen on stem cell differentiation towards preadipocytes depends on several factors including the cell-line origin, local hormone, and nutrient concentrations, as well as the presence of other molecules potentially interfering with the estrogen-related signaling pathways. The majority of these studies were carried out on rodent cell lines. In mouse bone marrow stromal cell line (ST-2), overexpressing Erα and Erβ, 17 β-E2 causes lineage shift towards osteoblasts [47]. However, in murine adipose-derived stromal/stem cells (ASCs), activation of ERα by its specific agonist propylpyrazoletriol (PPT) has been shown to stimulate, in a concentration-dependent manner, their differentiation towards adipocytes. In turn, activation of ERβ by its selective agonist diarylpropionitrile (DPN) was much less efficient in this aspect [48]. The exposition of human ASC to 17 β-E2 induced a pro-adipogenic differentiation reflected by the increased lipid vacuole formation and decreased alkaline phosphatase (ALP) activity. This ability of E2 to weaken ASC osteogenic potential was most evident in ASCs isolated from pre-menopausal women [49,50]. Experimental data suggest that the E2 effect on osteogenesis occurs via stimulation of both estrogen nuclear receptors α and β, whereas the effect on adipogenesis is ERα selective [51].

Valuable information on the role of estrogens in the regulation of adipogenesis was provided by animal studies with a knockout of key genes for the metabolism and action of these hormones, as well as clinical observations of patients affected by mutations within these genes. Mice with a global ERα-knockout (αERKO), regardless of sex, are characterized by a significant (50–180%) increase in adipocyte number that is accompanied by insulin resistance, glucose intolerance, and liver steatosis [52]. Similar phenotype is present in animals with an aromatase (cyp19) knockout (ArKO), which are unable to synthesize endogenous estrogens [53], and patients with mutations within CYP19A1 and ESR1 (encoding ERα) [54,55,56,57].

The chief mechanism by which estrogens influence adipocyte proliferation seems to be associated with the inhibition of peroxisome proliferator-activated receptor gamma (PPARγ) coactivators recruitment. These include, among others, steroid receptor coactivator-1 (SRC-1) and CREB-binding protein (CBP) [58]. Other potential pathways may involve the activation of cyclin-dependent kinase inhibitors (CDKIs) p27 and p21, since animals with a double p27 and p21 knockout present the same clinical phenotype as αERKO and ArKO mice [59]. Interestingly, mice with a deletion in ERβ (βERKO) are not characterized by increased adiposity, which suggests a major role of ERα in the regulation of adipocyte proliferation [60].

The impact of estrogen on ASC differentiation and proliferation is sex-specific. Phenotypic differences between αERKO and βERKO mice manifest after sexual maturation when sex steroid serum levels reach some kind of threshold [61]. Moreover, gender-related differences in estrogen local concentration contribute directly to the differential fat distribution between the sexes. While in female mice, HFD induces adipogenesis both in subcutaneous and visceral depots, in males-specifically in the visceral depot [7]. In turn, in clinical studies, subcutaneous and visceral preadipocytes from women were more responsive to E2 in stimulating proliferation than those originating from men. Interestingly, neither E1 nor dihydrotestosterone had a gender- or site-specific effect on the preadipocyte proliferation rate [62]. Subsequently, subcutaneous adipose tissue in healthy women was found to contain a higher content of early-differentiated adipocytes compared to men [63]. In addition, overnutrition has a gender-specific impact on the ASC proliferation rate, too. For instance, eight weeks of overfeeding in healthy women resulted in a more significant increase in the number of adipocytes in subcutaneous adipose tissue compared to men [63]. Of note, this phenomenon refers to healthy individuals and relatively short-time exposition to energy surplus. Otherwise, human obesity (regardless of gender) is associated with a decrease in the expression of both genes encoding ERs in adipose tissue, which can be restored after weight loss [38,39,40]. These findings suggest that obesity-related adipose tissue dysfunction, via downregulation of expression of genes encoding ERs, may guide the expansion of adipocytes by hypertrophy rather than hyperplasia.

Importantly, estrogen can also influence adipogenesis indirectly, via the regulation of critical steps of other steroid hormones synthesis. This refers to the ability of E2 to upregulate the activity of 11-β-hydroxysteroid dehydrogenase type 1 (11βHSD1) which converts inactive cortisone to active cortisol-an important adipogenesis upregulator in human preadipocytes [64]. Expression of 11βHSD1 correlates positively with CYP19A1 and ESR2 (encoding ERβ) mRNA levels in SAT from both premenopausal and postmenopausal women irrespective of the nutritional status, and with measures of central fat accumulation [65,66]. On the other hand, other steroid hormones present in the local milieu may influence the impact of estrogen on adipogenesis. This is the case with androgens whose administration to women results in the reduction of late-stage differentiation of pre-adipocytes to adipocytes [67]. This phenomenon seems to be responsible for the decreased lipid storage capacity in adipose tissue in women with hyperandrogenism and promotes lipotoxicity [67,68].

In addition, the physiological impact of estrogen on adipogenesis can be blunted via components known as endocrine disruptors, for instance, a xenoestrogen-Bisphenol A (BPA). BPA binds to estrogen receptors (ERα and ERβ) and subsequently, in a concentration-dependent manner, modulates the expression of several proadipogenic genes, including PPARγ, SRC-1, and CBP [69]. Exposition of human ASC to BPA results in the increased expression of genes encoding fatty acid synthase (FASN) and lipoprotein lipase (LPL) and subsequent, dose-dependent triglyceride accumulation [70]. However, epidemiological studies regarding the impact of prenatal exposure to BPA on birth weight have led to conflicting results [71,72].

### 3.2. Estrogens in the Regulation of White and Brown Adipogenesis

The main type of adipose tissue in the human body is WAT, and its various deposits may differ in storage capacity, as well as metabolic and secretory activity. Brown adipocytes present in humans can be of two different origins: part of them arises during embryogenesis (a constitutive brown adipose tissue-cBAT), while part develops postnatally within white adipose tissue depots and therefore is referred to as beige adipocytes or recruitable BAT (rBAT). Constitutive adipocytes are reached in mitochondria possessing uncoupling proteins 1 (UCP1) that can uncouple electron transport, and in this way-disperse energy cumulated in chemical bindings of adenosine triphosphate (ATP) as heat. Beige adipocytes are histologically very similar to cBAT and equivalent in their thermogenic potential; however, they express unique protein markers that correspond to their functional characteristics and allow to distinguish them from cBAT and white adipocytes. Irrespective of their origin, due to their metabolic properties, the induction of brown adipocytes constitutes an attractive therapeutic approach to combat obesity [1,73].

The results of the studies on how estrogen can influence ASCs’ fate are not unequivocal. In a study by Lapid et al., selective ERα activation induced murine ASCs towards white adipocyte lineage, while ERα deficiency reprogrammed the cells to differentiation towards smooth muscle or brown adipocytes. Subsequently, mice with a selective ERα knockout in adipose tissue (AT-ErαKO) were resistant to HFD and had reduced adipogenic potential and fat mass with increased energy expenditure and improved glucose sensitivity. The authors suggest that the underlying mechanism is probably based on ERα ability to inhibit tumor growth factor (TGF) β signaling, while ERα loss activates TGFβ expression and signal transduction, which reprograms progenitor cells into alternative fates, such as the smooth muscle lineage [74]. In turn, other authors confirmed a high potency of ERα in stimulating murine ASC proliferation and migration, while revealing that its activation induces expression of key genes for brown and beige adipocytes’ metabolism (e.g., UCP-1 and PPAR-γ), and, thus, may improve brown and beige adipogenesis. In contrast, induction of ERβ in murine ASCs repressed brown adipogenesis by decreasing the expression of these genes [48,75]. The ability of estrogen to induce BAT and increase its thermogenic activity was confirmed in animal models [76,77].

Several lines of evidence indicate that estrogen may potentiate the enhancement of BAT activation and beiging of WAT in humans. Imaging studies revealed that women have larger BAT depots and higher BAT thermogenic activity than men [78,79,80]. Moreover, women, compared to men, are characterized by increased recruitment and activation of BAT in response to cold [81,82,83]. Subsequently, BAT originating from women has a higher expression of mitochondrial genes, including UCP1 [84]. In turn, testosterone inhibits the thermogenic program in beige adipocytes [85]. Female adipose tissue has a higher sensitivity to activation of the sympathetic nervous system, a key mediator of BAT activation and/or beiging, too [86]. This finding is consistent with the results of preclinical studies indicating that estrogen has the potential for increasing BAT sympathetic nerve discharge and in this way-to stimulate BAT development and thermogenesis [87]. Sympathetic stimulation of brown and beige adipocytes results in lipolysis and subsequent production of heat and norepinephrine-stimulated lipolysis in women’s adipocytes exceeds that seen in men in vitro and in vivo [88]. However, the responsiveness of adipose tissue to lipolytic adrenergic stimulation depends on the depot and menopausal status [89]. Importantly, in humans, obesity is associated with a decrease in the expression of both estrogen receptor and thermogenesis-related genes in adipose tissue [39,90].

## 4. Estrogens in the Regulation of Adipose Tissue Metabolism

In addition to being involved in regulating ASC proliferation and partially also their fate, estrogens have been implicated in other adipocyte functions, including lipolysis, lipogenesis, insulin sensitivity, adipokine secretion, and immunoregulation. Lack of estrogen action due to their deficiency or inactivating mutations in ERs predisposes individuals to obesity and other components of metabolic syndrome.

### 4.1. Estrogens in the Regulation of Lipolysis/Lipogenesis

As it has been mentioned above, hypertrophy has been considered the main mechanism by which adipocytes react to energy surplus. The regulation of adipocyte size is linked to the balance between its lipolytic and lipogenic ability. Adipocyte’s lipid storage is determined by its capacity for fatty acid (FA) uptake and conversion to triacylglycerol (TAG)-the process named lipogenesis. TAG is stored in adipocytes in the form of lipid droplets and mobilized during lipolysis-the process catalyzed by lipoprotein lipase (LPL) leading to the breakdown of TAG into glycerol and free fatty acids (FFA) [91]. Among several factors determining adipocyte’s basal lipolytic activity (e.g., genetic variance, age, physical activity, and the location of the fat depot), sex seems to play a significant role, suggesting that sex steroids can regulate this dynamic and composed process [92]. Indeed, clinical studies show that in women the majority of circulating FAs is taken up by SAT, while in men, a significant number of FAs is preferentially stored in visceral fat [93].

Regulation of lipolysis/lipogenesis by estrogen may depend on distinct molecular mechanisms. ERs via interaction with an activator protein (AP-1)-like TGAATTC sequence located in the gene encoding lipoprotein lipase (LPL) downregulate its expression as it was shown in 3T3-L1 preadipocytes [94]. Moreover, in the mature murine white adipocytes, unlike in ASC, E2 was shown to decrease the expression of PPARγ and the PPARγ target genes implicated in lipogenic pathways [58]. However, in rodents, the involvement of estrogen in the regulation of lipolysis is not limited to the modulation of PPARγ activity. For instance, the ovariectomy-related decline in E2 level preserves the translocation of adipose triglyceride lipase (ATGL) to the lipid droplet and disturbs the phosphorylation of LIPE. Therefore, in rodents E2 deficiency seems to impair both basal and catecholamine-stimulated lipolysis, preferably in the visceral adipose tissue depot which can be reversed by estrogen replacement [95].

The results of studies on the impact of estrogen on lipolysis in human adipose tissue are univocal and depend on the studied model [92]. Pedersen et al. found that E2, via interaction with ERα and subsequent upregulation of antilipolytic α2 adrenergic receptors, attenuated catecholamine-stimulated lipolysis in primary subcutaneous adipocytes obtained from postmenopausal women, regardless of the use of hormone replacement therapy (HRT) [8]. In turn, in a study by Palin et al., high concentrations of E2 decreased, while lower concentrations were found to increase LPL expression in human primary subcutaneous adipocytes [96].

In general, women, whose visceral adipose tissue is characterized by a higher number of beta-adrenergic receptors, have increased sensitivity to catecholamine-induced lipolysis compared to men [97]. However, menopause and pharmacological castration in women lead to increased accumulation of VAT, which can be partially reversed by HRT [98,99]. Similarly, individuals lacking either aromatase or ERα tend towards visceral obesity, and estrogen treatment can improve their body composition, as well as ameliorate some of the metabolic complications, e.g., insulin resistance [55].

E2-based HRT was associated with a decrease in the expression of lipogenic genes such as fatty acid desaturase 1, acetyl CoA carboxylase alpha, stearoyl-coenzyme A (CoA) desaturase, and PPARγ in adipose tissue of post-menopausal women [100]. In turn, the impact of menopause on LPL activity is controversial: an initial study by Tchernof et al. indicated that adipose tissue of postmenopausal women is characterized by a higher LPL expression and basal lipolysis, compared to younger counterparts [101]. However, a subsequent work found no difference in LPL activity in VAT between pre- and postmenopausal women suggesting that increased adiposity in post-menopausal women may be due to the increased lipogenesis alone [102]. Concomitantly, three-month treatment with 2 mg estradiol valerate did not influence LIPE expression in SAT of postmenopausal women [103]. In contrast, in this study, administration of testosterone undecanoate (40 mg every second day) significantly downregulated LIPE protein level [104].

In rodents, the relationship between the estrogen status and lipolysis rate in SAT is even less evident-in some studies, ovariectomy led to a 50% decrease in lipolysis in subcutaneous adipocytes (restored by E2 supplementation) while in others, no difference of estrogen deprivation on lipolysis rate was observed [92,105]. These controversial findings in animal and human studies can be partially explained by the differences in local proportion between estrogen receptors [106]. The majority of signals regulating adipocyte metabolism seem to be mediated by ERα. Indeed, ERα-deficient mice have 100% more body fat compared to wild-type animals. The consequence of ERα knockout is the accumulation of 17β-E2 and the activation of ERβ signaling. However, ovariectomy in αERKO mice, leading to a loss of ERβ signaling, results in a decrease in body weight and fat deposits, which correlates with improved insulin sensitivity and carbohydrate metabolism. Thus, it appears that the ERβ-mediated effects of estrogens are opposite to those mediated by ERα [107].

Moreover, the impact of estrogens on the lipolysis and lipogenesis pathways can be modulated by locally and systemically acting androgens, which can be observed, for example, during the menstrual cycle. Higher androgen levels dampen lipolysis and LIPE activity in VAT in the luteal phase of the ovarian cycle, while in the follicular phase, testosterone has been found to increase fatty acid uptake. In healthy women, androgens, via inhibition of lipolysis and the stimulation of lipogenesis, favor the accumulation of VAT [92,108]. Thus, the relative balance of estrogen and androgen may be crucial for the regulation of lipolysis and lipid storage in adipose tissue.

In summary, multiple factors, such as the source (depot) of the investigated tissues, local androgen concentrations and proportions between ERα and ERβ, and the type of estrogen used could contribute to the described above discrepancies. Further studies focused on understanding sex-steroid and gene interplay and observational studies to describe differences in male and female WAT adipose tissue distribution and activity are necessary to resolve these associations.

### 4.2. Estrogens in the Regulation of Adipose Tissue Insulin Sensitivity

Adipose tissue, along with the liver and skeletal muscle, is a key organ involved in the regulation of glucose metabolism and insulin sensitivity [109]. Insulin resistance is defined as the inability of insulin to effectively regulate the uptake and/or utilization of glucose by insulin-sensitive tissues and organs. In insulin-sensitive individuals, hyperglycemia stimulates insulin release from pancreatic β-cells and inhibits gluconeogenesis in the liver. However, in an insulin resistance state, insulin secreted in response to hyperglycemia is neither able to stimulate glucose uptake in peripheral tissues (including adipose tissue), nor inhibit glucose production in the liver [110].

In clinical studies, premenopausal women when compared to women after menopause and the respective age-matched men, are characterized by higher insulin sensitivity assessed by homeostatic model assessment–insulin resistance (HOMA-IR) and adipose-insulin resistance index [111]. Both age-related and surgically induced menopause increase insulin resistance and the prevalence of other components of the metabolic syndrome [112]. These findings suggest the role of estrogen in the regulation of responsiveness to insulin [113]. However, the results of the studies evaluating the influence of HRT on glucose metabolism are not univocal which may be a consequence of methodological differences regarding the population selection, HRT type, and regimen as well as the timing of treatment initiation [114].

Nevertheless, meta-analyses point to the protective effect of exogenous estrogens on the risk of insulin resistance and diabetes onset [13,115]. This insulin-sensitizing effect of estrogen is related to its influence on adipose tissue distribution: estrogen prevents the accumulation of visceral abdominal fat in female mice and protects them from developing insulin resistance [116]. Of note, studies in men with either estrogen deficiency due to the CYP19A1 mutation or estrogen resistance due to a mutation in ESR1 indicate that estrogens are crucial for the regulation of insulin sensitivity and glucose metabolism in both sexes [54,117]. This is different from testosterone which plays a key role in the regulation of glucose and lipid metabolism, too, but in different manners in men and women. In men, low concentrations of testosterone are associated with obesity and adipose tissue insulin resistance, while in women, excess testosterone has an unfavorable effect on insulin sensitivity [118].

### 4.3. Estrogens in the Regulation of Adipokines Secretion

The identification of leptin and adiponectin about 30 years ago changed our concept of adipose tissue function. It turned out that apart from storing energy, it is capable of secreting several mediators acting in an auto-, para-, and endocrine manner. These substances-adipokines-have a significant impact on the functioning of other organs and tissues. Subsequently, obesity-associated adipose tissue dysfunction contributes to an unfavorable change in the profile of substances secreted by adipocytes, and thus-to the development of obesity-related complications [1]. Estrogens, by regulating adipocyte gene expression, play an important role in the control of substances secreted by adipose tissue.

Obesity is related to an increase in the secretion of leptin-a key adipokine for maintaining the body’s energy homeostasis (stimulating satiety and tissue sensitivity to insulin), which also, among others, regulates the tone of the vascular walls and reproductive functions. However, obesity-related peripheral tissue resistance to leptin makes the hormone unable to exert its beneficial metabolic and cardiovascular effects [119].

In vitro studies suggest that leptin synthesis and secretion are controlled by sex steroids; however, the effect depends on the intracellular Erα/Erβ ratio, since the interaction of estrogen with Erα in 3T3-L1 adipocytes induces leptin expression, binding with Erβ exerts an opposite effect [120]. Moreover, studies on primary human omental adipocytes suggest that the effect of estrogen on leptin synthesis can be gender specific: in postmenopausal women, E2 increases expression of the LEP gene, but in cells derived from age-matched men, it does not [121]. Surprisingly, both αERKO and ArKO mice have elevated serum leptin levels; however, it can be an effect of increased adiposity itself, rather than the consequence of the lack of estrogen signaling, especially since both animal and human studies suggest that estrogen treatment results in an increase in serum leptin levels [52,122,123,124]. This finding is in agreement with the fact that women, whose estrogen levels are naturally elevated compared to men, also have higher serum leptin concentrations. Moreover, a menopause-related decline in estrogen levels is associated with a decrease in serum leptin, unless a woman gains weight-then increased adiposity leads to hyperleptinemia [125]. However, in normal-weight subjects, the SAT of premenopausal women is characterized by higher LEP mRNA levels compared to postmenopausal ones, due to the higher systemic and local E2 levels [125]. However, neither administration of estrogen-based oral contraception nor HRT had an influence on serum leptin levels after adjustment for the body mass index (BMI) [126].

Obesity is associated with a decrease in the concentration of anti-inflammatory and antioxidant adiponectin, which correlates with an increased risk of heart and vascular diseases. From the point of view of the metabolic risk of obese patients, the adiponectin/leptin ratio seems to be of key importance, the value of which decreases with increasing insulin resistance [127]. Even though adiponectin levels do not differ concerning the menopausal status and in animal studies, ovariectomy did not alter plasma adiponectin concentrations, they are lower in age- and BMI-matched males than females. The gender-specific difference occurs during puberty and correlates with changes in serum testosterone levels [128]. This finding is consistent with in vitro studies, where testosterone reduced adiponectin secretion in murine 3T3-L1 adipocytes, while high levels of plasma adiponectin were found in castrated mice and restored to those observed in the control animals after testosterone treatment [129]. The results of studies on the impact of ovariectomy on adiponectin levels in rodents are ambiguous. Ovariectomized rats, despite increased visceral adiposity, do not demonstrate changes in adiponectin levels, and their treatment with 17 β-E2, even though it reduces fat mass and improves metabolic health, has no influence on adiponectin level [130]. However, in mice, ovariectomy resulted in a decrease in adiponectin serum concentration, which was restored by the treatment with exogenous estrogens [122]. Accordingly, 17β-E2 can suppress the expression of the adiponectin gene in murine 3T3-L1 preadipocytes, but not in the human SBGS adipocyte cell line [131,132]. Concomitantly, in mice, neonatal castration allowed adiponectin levels to reach female adult levels, suggesting that androgens rather than estrogens act as a crucial regulator of its expression [133]. Adiponectin is present in the circulation in low, medium, and high molecular weight (HMW) oligomeric forms and the ratio of HMW forms of total adiponectin seems to determine its ability to regulate insulin sensitivity and cardiovascular properties. Androgens have an impact on the serum ratio of adiponectin oligomeric complexes. The concentration of HMW adiponectin forms was found to be significantly higher in female mice than in male subjects. Subsequently, castration induces an elevation of the HMW form in sera in experimental animals. In agreement, hypogonadal men have higher HMW adiponectin levels compared to non-hypogonadal individuals, which can be restored by testosterone supplementation. In vitro studies confirmed a direct selective inhibition of HMW adiponectin formation by testosterone [134]. This specific reduction in adiponectin HMW could constitute a potential mechanism for a higher prevalence of cardiometabolic diseases in males than females. Notably, in humans, an obesity-related decrease in ER concentration in adipose tissue is accompanied by a decline in adiponectin mRNA and protein levels [39,135].

The adipose tissue of obese individuals secretes lower amounts of omentin-1-an adipokine of properties similar to that of adiponectin whose serum concentration negatively correlates with cardiometabolic risk. In experimental animals, ovariectomy leads to a decrease in omentin-1 serum level which is accompanied by an increase in serum glucose and insulin levels and can be restored by estrogen replacement therapy [136]. Data on the influence of estrogen on omentin-1 levels in humans are scarce; however, in patients with prostate cancer, circulating omentin-1 concentrations correlated neither with estradiol nor testosterone levels, suggesting that the regulatory mechanism is different than in the case of adiponectin [137].

Resistin is an example of another adipokine in which serum levels are altered in the course of obesity. Resistin is secreted by macrophages infiltrating adipose tissue and by activating the nuclear factor κB (NF-κB), increases the expression of pro-inflammatory cytokines, chemokines, and adhesion molecules that attract more immunocompetent cells to the adipose tissue, as well as endothelin-1 involved in the process of endothelial dysfunction [138]. Finding that resistin serum levels differ, in a species-specific manner, between genders both in rodents and in humans suggests a possible involvement of sex steroids in regulating RSTN gene expression [139]. Estrogen-responsive elements (ERE) have been identified in the regulatory region of RSTN, and E2 was found to upregulate resistin expression in murine 3T3-L1 [131,140]. However, in ovariectomized mice, 17 β-E2 replacement was found to decrease RSTN expression in adipose tissue [122,141]. Data on the impact of estrogen on resistin secretion in adipose tissue are scarce; however, in obese patients, an opposite trend in the expression of genes encoding ERs and RSTN in adipose tissue can be observed [39,135].

Visfatin secreted by adipocytes and macrophages has a similar profile of action to resistin. This adipokine is preferentially expressed in VAT as compared with SAT and its plasma level correlates positively with the volume of visceral fat depots in humans and increased cardio-metabolic risk, as well as the risk of all-cause mortality. As in the case of resistin, treatment of murine 3T3-L1 cells with estrogens leads to an increase in visfatin gene expression. However, estradiol is much less potent in this aspect than estriol, pointing at the particular role of this regulatory pathway in pregnancy [142]. Interestingly, the interaction between visfatin and ERs can be bidirectional. In human breast cancer line MCF-7, administration of visfatin, through MAPK and PI3K/Akt signaling pathways, increases the phosphorylation of ERα at serine 118 (Ser118) and 167 (Ser167) residues in vitro and enhances ERE-dependent activity of ER in the presence of 17-β estradiol (E2) [143].

Undoubtedly, estrogens play an important role in regulating adipose tissue secretory activity. The above-described discrepancies between the studies investigating the effect of estrogen on adipokine expression may result from the differences in the experimental design: the investigated cell line, species, type of estrogen, and its concentration. Moreover, local, depot-dependent differences in ER subtypes expression could have impacted the results.

### 4.4. Estrogens in the Regulation of Metabolic Inflammation

Overloading adipocytes with lipids causes several changes in their functioning, including mitochondrial dysfunction and endoplasmic reticulum (ER) stress, leading to hypoxia, fibrosis, and, consequently, cell death. As a result, the expression of pro-inflammatory genes in adipocytes increases, which in turn attracts infiltrating immune cells. The latter secrete their own inflammatory mediators, intensifying adipocyte dysfunction. Therefore, obesity is often accompanied by a chronic inflammatory state, called metainflammation, which correlates with the occurrence of cardiometabolic complications of obesity [144].

In normal-weight individuals, M2 macrophages present in adipose tissue promote anti-inflammatory signals. However, obesity-related changes in adipocyte secretory profile (including, e.g., excess synthesis of interleukin 6 (IL-6) and tumor necrosis factor-alpha (TNFα)) attract pro-inflammatory M1 macrophages to perpetuate pro-inflammatory cytokine signaling to trigger adipocyte cell death [145]. There are several lines of evidence suggesting that estrogen can suppress pro-inflammatory signals in adipose tissue; however, their immunomodulatory effect may depend on bioavailability, concentration, immune cell type, immune stimulus, and ER subtype expression [146].

Macrophages devoid of ERα are unable to respond appropriately to typical stimuli (e.g., liposaccharide), which results in their diminished ability for phagocytosis and changes in secretory activity (increased secretion of IL-1b, IL-6, and interferon γ). Notably, in mice with a selective ERα knockout in hematopoietic/myeloid cells, these alternations are accompanied by increased insulin, leptin, and plasminogen activator inhibitor-1 (PAI-1) levels and translate to increased insulin resistance, glucose intolerance, and accelerated formation of atherosclerotic plaques. This last phenomenon may result from the fact that estradiol is a transcriptional regulator of transglutaminase (Tgm) 2, an enzyme protective against atherosclerotic lesion development [147].

In agreement, αERKO mice are characterized not only by increased intra-abdominal adipose tissue mass and increased adipocyte size, but also by elevated adipose tissue inflammation (assessed by the expression of genes encoding IL-6 and TNFα) regardless of gender. Interestingly, a selective knockdown of adipocyte ERα in the context of the βERKO mouse also increases inflammation and fibrosis, indicating a role for ERβ in the absence of adipocyte ERα [148]. Similarly, ArKO mice have increased systemic IL-6 and TNFα concentrations and recruitment of the pro-inflammatory M1 macrophages [53], while aromatase overexpression (but not 17 β-E2 treatment) reduced adipose tissue inflammation [61,123]. The opposite trend between the expression of genes encoding ERs and proinflammatory cytokines (namely IL-1β and IL-6) has also been observed in adipose tissues of obese individuals [39,44]. Of note, in other experimental settings, treatment with estradiol successfully prevented LPS/IFN-γ stimulation of human M2 macrophages [149].

In humans, menopause or ovariectomy leads to an increase in the proinflammatory cytokine serum levels. Postmenopausal women, compared to premenopausal ones, have higher leukocyte counts and higher plasma TNF-α, IL-1β, and IL-6 levels [150]. Moreover, menopausal estrogen loss is associated with an increased number of senescent T-cells and with a blunted activation of M2 macrophages leading to an increased M1/M2 response ratio, which may affect the cardiovascular risk profile concerning menopausal status [149,150]. However, meta-analyses suggest that properly chosen HRT reduces the proinflammatory state in postmenopausal women with the metabolic syndrome, parallel to the decrease in insulin resistance, fasting glucose, and new-onset diabetes in women without diabetes and reduced insulin resistance and fasting glucose in women with diabetes [13]. Importantly, given the anti-inflammatory properties of androgens, the local estrogen-to-androgen ratio may act as a crucial modulator of metabolic inflammation [151].

The findings from studies on the role of estrogens in the regulation of adipogenesis and adipose tissue activity in various experimental settings are summarized in Table 1.

## 5. Therapeutic Potential of Estrogen in the Treatment of Obesity

Given the significant role of estrogen in the regulation of adipocytes’ development and metabolism, estrogen receptor modulators may be an option for the treatment of obesity and its complications.

In epidemiological studies, a menopause-associated decline in estrogen levels correlates with an increased prevalence of obesity in women and unfavorable changes in adipose tissue distribution, which can be partially prevented by hormone replacement therapy [11,12,15]. Since this effect is more pronounced in normal-weight subjects, HRT can be considered rather as a preventive measure than a therapeutic strategy and is not included in obesity-management algorithms. The challenge with estrogen therapy, however, is its narrow therapeutic index when administered as a chronic treatment. Therefore, the administration of exogenous estrogen or selective ER modulators requires regular screening for oncological complications [152].

Since HRT may raise some safety concerns, natural ER modulators may be a therapeutic option in the treatment of obesity [153]. An example of such compounds are phytoestrogens present in plant products, which resemble human estrogens in terms of their chemical structure and biological functions. Given their chemical structure, phytoestrogens can be divided into two groups: flavonoids (which include soy isoflavones and coumestans) and non-flavonoids (lignans and resorcinol derivatives).

Phytoestrogens have shown estrogen-like effects on adipogenesis and adipocyte metabolism in vitro and in animal models of obesity. However, the influence of phytoestrogens consumption and the occurrence of obesity in humans is not fully clear. On the one hand, epidemiological studies suggest their protective effect. For example, it has been shown that in women, a higher lignan concentration in the urine correlates with an approximately 50% lower probability of obesity, especially visceral (National Health and Nutrition Examination Survey (NHANES) 2003–2008) [154]. In addition, in the NHANES 2001–2010 study, a higher lignan content in the urine was associated with a lower risk of developing metabolic syndrome [155]. On the other hand, the results of randomized controlled trials (RCT) evaluating the impact of phytoestrogen intake on body weight are not so clear and suggest that the effect of phytoestrogens on body weight is compound-specific and depends on metabolic status. For example, the use of supplements containing a mixture of isoflavones was associated with weight loss in healthy postmenopausal women, while the use of daidzein alone resulted in weight gain in women with metabolic syndrome [156].

Moreover, the intestinal microflora responsible for the transformation of dietary phytoestrogens into active metabolites can modulate the effectiveness of phytoestrogens supplementation [157]. Therefore, further well-designed RCTs are needed to evaluate the therapeutic potential of phytoestrogens in the treatment of obesity in humans [158].

Therapeutic hopes are also associated with compounds selectively modulating the activity of estrogen receptors (SERMs), which could enhance the metabolic actions of estrogen in chronic therapies aiming at the prevention of metabolic dysfunction in postmenopausal women. SERMs interact with estrogen receptors (ERs) as ligands inducing their conformational changes and leading to various responses in estrogen-sensitive tissues. SERMs may have agonist or antagonist properties, depending on their structure, but also target tissue, local ERα/ERβ ratio, and availability of coactivators and corepressors [20]. In addition to their effects on the breast, bone, and endometrium (described in detail elsewhere), SERMs have an impact on metabolic homeostasis [20].

Tamoxifen is a representative of the first generation of SERMs, acting as an ER antagonist in breast tissue and as an agonist in the uterine endometrium increasing endometrial carcinoma risk. Tamoxifen is routinely used in the therapy of ER-positive breast cancers; however, obese women without breast cancer taking tamoxifen note lower weight gain (assessed by BMI increase) compared to those taking a placebo. The anorectic impact of tamoxifen is related to its ability to influence neuronal transmission in the hypothalamus [159]. Nevertheless, the impact of tamoxifen on metabolic health is not so unequivocal. In premenopausal, normal-weight women, tamoxifen did not have an impact on glucose metabolism; however, in an overweight subgroup, its administration dramatically increased insulin resistance leading to the development of diabetes [160]. The underlying mechanisms are not fully elucidated but may be related to the ability of tamoxifen to promote liver steatosis [161]. In addition, tamoxifen treatment is associated with an increased risk of hypertriglyceridemia [162].

Raloxifene belongs to the second-generation SERMs, devoid of the disadvantages of tamoxifen, showing anti-estrogenic activity in breast tissue and estrogenic in bones. Clinical studies suggest that raloxifene administration has a favorable influence on body composition. One-year treatment with raloxifene promoted adipose tissue redistribution from android to gynoid and prevented weight gain in healthy postmenopausal women [163]. Even though raloxifene does not decrease body weight directly, it increases fat-free mass that transfers to beneficial cardiometabolic outcomes [164,165]. Indeed, raloxifene treatment has neither adverse impact on glucose metabolism nor triglyceride level and significantly decreased total and low-density-lipoprotein (LDL) cholesterol levels with a parallel increase of high-density lipoprotein (HDL) cholesterol that was confirmed by the recent meta-analysis [166,167].

Bazedoxifene (BZA) is a third-generation SERM that exhibits estrogen antagonistic activity in the breast and uterus while it has a similar effect on bone as raloxifene and, combined with conjugated estrogen (CE/BZA), is approved for menopausal therapy [168]. Pooled analysis from five randomized, double-blind, placebo- and active-controlled studies in postmenopausal women showed that CE/BZA administration prevents menopause-related weight gain [169]. Moreover, this treatment does not influence glucose metabolism and has a favorable effect on lipid profile (reduces LDL and increases HDL cholesterol level) [170]. It seems that SERMs with properties similar to BZA and targeting only the ER involved in energy balance may be a therapeutic option for preventing menopausal weight gain and metabolic disorders.

In general, modulation of steroid hormone activity to combat metabolic dysfunctions and weight gain represents a therapeutic strategy that develops rapidly during the last years. Selective androgen receptor modulators (SARMs) depending on their chemical structure can act as agonists, antagonists, partial agonists, or partial antagonists of the androgen receptors within different tissues [171]. Therefore, the administration of SARMs results in anabolic effects without unfavorable side effects typical for anabolic steroids. In metabolic aspects, SARMs have been studied in the treatment of cancer-associated cachexia due to their ability to increase fat-free mass [172].

## 6. Conclusions

Several lines of evidence point to the significant role of estrogen in the regulation of adipose tissue development and function in mammals. Therefore, disturbances in estrogen availability and/or function may lead to increased adiposity and metabolic complications resulting from impaired lipolysis, adipokine secretion, as well as altered immune responses. Up to now, the majority of data on estrogens’ role in obesity pathogenesis come from preclinical studies with cells or animals devoid of estrogen. However, their results are convergent in many points with clinical trials analyzing the impact of menopause-associated estrogen decline or hypogonadism on body composition and metabolic status.

Data on the impact of obesity on estrogen metabolism and action in humans are scarce. Nevertheless, available studies suggest that excess adiposity is associated with decreased expression of ERs and key enzymes involved in estrogen synthesis [38,39]. Given the inhibitory effect of estrogens on adipogenesis and lipogenesis, it can be concluded that their deficiency or impaired function may predispose to the development of obesity. However, obese individuals have reduced expression of ERs in the adipose tissue [40]. One might wonder what the primary mechanism is: whether the decrease in the expression of key genes involved in the metabolism and action of estrogens is the result or the cause of obesity. The finding that surgically induced weight loss restores the expression of these genes to levels seen in normal-weight individuals suggests that the impaired estrogen activity in adipose tissues of obese patients is secondary to weight gain. On the other hand, as the menopausal status may impact the activity of genes in adipose tissue, a menopause-related decrease in the expression of genes crucial for estrogen synthesis and action may predispose postmenopausal women to weight gain and metabolic disturbances.

To date, the therapeutic use of estrogens and ERs-modulating compounds alone in the treatment of obesity is limited. HRT should be seen as an element of the prevention of cardiometabolic diseases in postmenopausal women rather than as a method of treating existing conditions. However, it cannot be ruled out that compounds with estrogenic activity may potentiate the effect of other drugs with proven beneficial effects on cardiometabolic complications of obesity, such as sodium-glucose transporter type 2 (SGLT2i) inhibitors [173]. However, this promising approach requires support coming from preclinical and clinical trials.

## Figures and Tables

**Figure 1 biomedicines-11-00690-f001:**
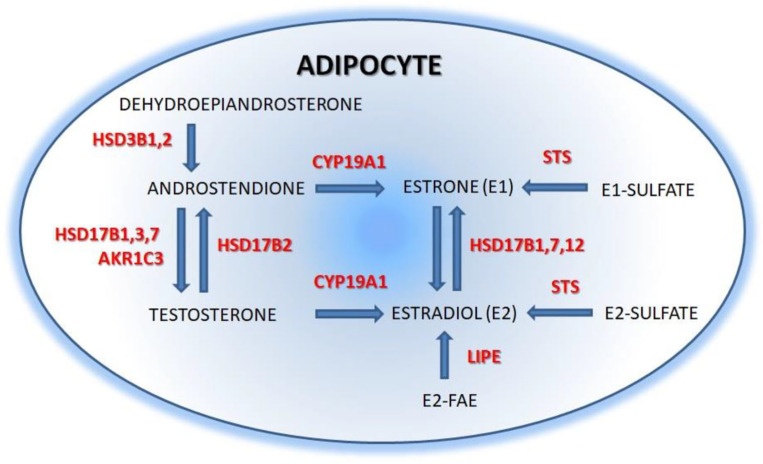
Synthesis of sex steroids in adipocyte (a simplified version). AKR1C3-aldo-keto reductase family 1 member C3 (17β-hydroxysteroid dehydrogenase type 5); CYP19A1-aromatase; E2-FAE-E2 fatty acyl esters; HSD3B1,2-3β-hydroxysteroid dehydrogenase types 1 and 2; HSD17B1,3,7,12-17β-hydroxysteroid dehydrogenase types 1, 3, 7 & 12; LIPE-hormone-sensitive lipase, STS-steroid sulfatase.

**Table 1 biomedicines-11-00690-t001:** Role of estrogens in the regulation of adipogenesis and adipose tissue activity.

Process/Function	Compound	Experimental Settings	Effect	References
**Stem cell differentiation**	17 β-E2	mouse BMSC	↑ differentiation towards osteoblasts	[47]
	propylpyrazoletriol (ERα agonist)	murine ASC	↑↑↑ differentiation towards adipocytes	[48]
	diarylpropionitrile (ERβ agonist)	murine ASC	↑ differentiation towards adipocytes	[48]
	17 β-E2	human ASC	↑↑↑ differentiation towards adipocytes	[49,50]
**Preadipocyte proliferation**	17 β-E2	subcutaneous and visceral preadipocytes	↑↑↑ proliferation in women ↑ proliferation in men	[62]
	E1	subcutaneous and visceral preadipocytes	↑ proliferation in women and men	[62]
**Adipocyte** **browning/beiging**	propylpyrazoletriol (ERα agonist)	3T3-L1 murine preadipocytes	↑ expression of beiging markers	[75]
	propylpyrazoletriol (ERα agonist)	murine primary preadipocytes	↑ expression of beiging markers	[75]
	17 β-E2	ovariectomized rats	↑ expression of browning markers in adipose tissue	[76]
	17 β-E2	ovariectomized rats	↑ expression of browning markers in adipose tissue	[77]
**Lipolysis/lipogenesis**	17 β-E2	3T3-L1 murine preadipocytes	↓ expression of lipoprotein lipase gene	[94]
	17 β-E2	mature murine white adipocytes	↓ expression of PPARγ gene (↓ lipogenesis)	[58]
	17 β-E2	human primary subcutaneous adipocytes (pre-and postmenopausal women)	↓ activity of lipoprotein lipase and hormone-sensitive lipase	[8]
	17 β-E2 high concentrations	human primary subcutaneous adipocytes (pre-and postmenopausal women)	↓ expression of lipoprotein lipase gene	[96]
	17 β-E2 low concentrations	human primary subcutaneous adipocytes	↑ expression of lipoprotein lipase gene	[96]
	17 β-E2	human subcutaneous adipose tissue samples (postmenopausal women)	↓ expression of stearoyl-CoA desaturase, acetyl CoA carboxylase alpha, fatty acid desaturase, PPARγ genes	[100]
	17 β-E2	human subcutaneous adipose tissue samples (postmenopausal women)	=expression of hormone-sensitive lipase gene	[103]
**Insulin sensitivity**	HRT	meta-analysis of studies in postmenopausal women	↓ insulin resistance ↓ diabetes onset	[13,115]
	17 β-E2	ovariectomized C57BL/6 mice	↑ insulin sensitivity	[116]
**Adipokine secretion**				
Leptin	propylpyrazoletriol (ERα agonist)	3T3-L1 murine preadipocytes	↑ expression of leptin gene	[120]
	diarylpropionitrile (ERβ agonist)	3T3-L1 murine preadipocytes	↓ expression of leptin gene	[120]
	17 β-E2	primary human omental adipocytes (postmenopausal women)	↑ expression of leptin gene	[121]
	17 β-E2	primary human omental adipocytes (men)	↓ expression of leptin gene	[121]
	17 β-E2	ovariectomized mice	↑ leptin gene expression in adipose tissue	[122]
	HRT	postmenopausal women	=serum leptin levels	[126]
	ethinyl estradiol	premenopausal women	=serum leptin levels	[126]
Adiponectin	17 β-E2	3T3-L1 murine preadipocytes	↓ adiponectin secretion	[131]
	17 β-E2	human SBGS adipocyte cell line	=adiponectin secretion	[132]
	17 β-E2	ovariectomized rats	=serum adiponectin levels	[130]
	17 β-E2	ovariectomized mice	↑ adiponectin gene expression in adipose tissue	[122]
Omentin	17 β-E2	ovariectomized rats	↑ omentin serum level	[136]
Resitin	17 β-E2	3T3-L1 murine preadipocytes	↑ resistin secretion	[131,140]
	17 β-E2	ovariectomized mice	↓ resistin gene expression in adipose tissue	[122,141]
Visfatin	17 β-E2	3T3-L1 murine preadipocytes	↑ visfatin gene expression	[142]
	E3	3T3-L1 murine preadipocytes	↑↑↑ visfatin gene expression	[142]
**Metabolic inflammation**	17 β-E2	female aromatase knockout mice	↑ IL-6 and TNFα serum level	[123]
	HRT	meta-analysis of studies in postmenopausal women	↓ serum levels of proinflammatory cytokines	[13]

17 β-E2-17β estradiol; ASC-adipose-derived stromal/stem cells; BMSC-bone marrow stromal cells; E1-estrone; E3-estriol; ER-estrogen receptor; HRT-hormone replacement therapy; IL-6-interleukin 6; PPARγ-peroxisome proliferator-activated receptor gamma; TNFα-tumor necrosis factor-alpha; ↑—increase; ↓—decrease.

## Data Availability

Not applicable.

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
