# Peer review of "Estrogens in Adipose Tissue Physiology and Obesity-Related Dysfunction"

_biomedicines, 2023, doi:10.3390/biomedicines11030690_

Round 1

Reviewer 1 Report

This is my review on estrogen in Adipose Tissue Physiology and Obesity-Related Dysfunction. 

This review aims to study the role of estrogen in adipose tissue development, physiology, and obesity.

The first paragraph, the introduction, is written well. The second paragraph regarding the estrogens’ synthesis and mechanisms needs to be rewritten in a more concise way in order to keep interesting to the readers. Figure 1 is fine. The rest paragraphs present interestingly the multifactorial role of estrogens in adipose tissue regarding obesity dysfunction. This is a well-presented review. My suggestion to the author would be to be concise in several paragraphs in order to avoid long book-like texts to make it even more interesting. 

Author Response

Reviewer 1

This is my review on estrogen in Adipose Tissue Physiology and Obesity-Related Dysfunction. 

This review aims to study the role of estrogen in adipose tissue development, physiology, and obesity.

The first paragraph, the introduction, is written well. The second paragraph regarding the estrogens’ synthesis and mechanisms needs to be rewritten in a more concise way in order to keep interesting to the readers. Figure 1 is fine. The rest paragraphs present interestingly the multifactorial role of estrogens in adipose tissue regarding obesity dysfunction. This is a well-presented review. My suggestion to the author would be to be concise in several paragraphs in order to avoid long book-like texts to make it even more interesting.” 

I thank the Reviewer for the positive reception of our manuscript and for drawing attention to some weak points, I tried to improve.

The second paragraph regarding the estrogens’ synthesis and mechanisms needs to be rewritten in a more concise way in order to keep interesting to the readers.”

Following the Reviewer's valuable suggestion the above-mentioned sections were modified to make them more concise.

“The key enzyme for estrogen synthesis is aromatase (CYP19A1) whose expression and activity have been reported in many human tissues including, e.g. endometrium, brain, bone, skin, and adipose tissue [25,26]. The effect of aromatase activity depends on the local availability of its substrates – androgens (Figure 1). For instance, in the ovary, where the main available androgen is testosterone, CYP19A1 activity results in the synthesis of estradiol. On contrary, in the adipose tissue, the main aromatase’ substrate is androstenedione (delivered from the dehydroepiandrosterone (DHEA) and its sulfate both synthesized in adrenals) which aromatization leads to the synthesis of estrone [24]. The conversion rate of androstenedione into estrone increases with age and adipose tissue volume, and it is higher in women with gynoid than in those with android obesity [27]. Estrone can also be synthesized in the adipose tissue by the oxidation of estradiol in the reaction catalyzed by 17β-hydroxysteroid dehydrogenases (17β-HSD) types 1, 7, and 12. Subsequently, estrone can be converted by steroid sulfotransferase (STS) to estrone sulfate, which is the most significant component of the pool of circulating estrogen. Another enzyme essential for the local availability of sex steroids is hormone-sensitive lipase (LIPE) which hydrolyses fatty acyl esters (FAE) of DHEA and E2 [24,28]. Fatty acyl esterified E2 is its storage form, unable to exert its biological functions. Therefore E2 esterification/hydrolysis balance is an important regulatory mechanism of biologically active steroid levels [29,30]. Estrogens are eliminated from the body mainly as sulfated and glucuronidated derivatives. The first step in this process involves the generation of hydroxylated derivatives. The hydroxyl group can then be sulfated, glucuronidated, or methylated [31].” Pages 2-3, Lines 94-115.

“Estrogen exerts its biological functions via interaction with its receptors (ERs) which can be both: nuclear and membrane-associated. Nuclear ERs exist in two main forms, α and β, which differ in their tissue expression and function [35]. While ERα plays a stronger physiologic role in females, the activity of ERβ is similar in men and women [36]. Upon ligand binding, ERs undergo conformational changes that allow for the formation of heterodimers and interaction with estrogen response elements (ERE) in the promoter of a target gene[37]. However, ERs can also act in an ERE-independent manner by modulation of co-regulatory proteins and transcription factors that are bound to their cognate responsive elements on DNA. Obesity is associated with a significant decrease in the expression of both nuclear ER subtypes in adipose tissue, while weight loss leads to an increase in ERα and ERβ mRNA levels [38,39]. Importantly, since ERs subtypes have a diverse impact on gene transcription the local ERα/ERβ ratio is critical for the final effect of estrogen action in a particular tissue [9]. The proportion between ERα and ERβ in adipose tissue can physiologically evolve with ageing, but also be disturbed by pathological conditions [40]. This is the case with obesity: adipose tissue of obese individuals of both sexes is characterized by a higher ERα/ERβ ratio compared to the tissues obtained from the normal weight subjects [39,40].

Additionally, estrogen can act rapidly (in a non-classical or non-genomic mechanism) via membrane-associated receptors interacting with other signaling molecules, e.g. G proteins, growth factor receptors, tyrosine kinases (Src), etc. The most prominent form of membrane ER is GPER1 (G protein-coupled membrane receptor1)[41,42].

In turn in the mitochondria, ERs can modulate gene expression either by direct interaction with the mtDNA or by increasing the activity of manganese superoxide dismutase. In addition, the activity of mitochondria can be modulated by the nuclear ERs that regulate the expression of genes crucial for mitochondrial actions [41,43].” Page 4, Lines 134-158.

My suggestion to the author would be to be concise in several paragraphs in order to avoid long book-like texts to make it even more interesting.” 

Following the Reviewer’s suggestion several paragraphs within the text were rearranged, and all changes are marked with green in the revised version of the manuscript.

Reviewer 2 Report

My compliments with the author, one stand person for this review. But i believe that some references can be added regarding obesity and oxidative stress as the new role of SGLT2 inhibitors in the management of heart failure: current evidence and future prespective. The introduction must be amplified adding also the presence of insulin resistence in obesity.

Author Response

Reviewer 2:

“My compliments with the author, one stand person for this review. But I believe that some references can be added regarding obesity and oxidative stress as the new role of SGLT2 inhibitors in the management of heart failure: current evidence and future perspective. The introduction must be amplified adding also the presence of insulin resistance in obesity.”

I thank the Reviewer for the positive reception of our manuscript and for drawing attention to some weak points, I tried to improve.

But I believe that some references can be added regarding obesity and oxidative stress as the new role of SGLT2 inhibitors in the management of heart failure: current evidence and future perspective."

Following the Reviewer’s suggestion the following paragraph with a proper reference was added to the Conclusions section in the revised version of the manuscript.

“To date, the therapeutic use of estrogens and ERs-modulating compounds alone in the treatment of obesity is limited. HRT should be seen as an element of the prevention of cardiometabolic diseases in postmenopausal women rather than as a method of treating existing conditions. However, it cannot be ruled out that compounds with estrogenic activity may potentiate the effect of other drugs with proven beneficial effects on cardiometabolic complications of obesity, such as sodium-glucose transporter type 2 (SGLT2i) inhibitors [175]. However, this promising approach requires support coming from preclinical and clinical trials.” Page 17, Lines 699-706.

[175] Muscoli, S.; Barillà, F.; Tajmir, R., Meloni, M.; Della Morte, D.; Bellia, A.; Di Daniele, N.; Lauro, D.; Andreadi, A. The New Role of SGLT2 Inhibitors in the Management of Heart Failure: Current Evidence and Future Perspective. Pharmaceutics 202214, 1730.

The introduction must be amplified adding also the presence of insulin resistance in obesity.

Following the Reviewer’s valuable suggestion, the following changes were introduced:

“This process, known as adipose tissue dysfunction, is believed to underlie the development of insulin resistance predisposing to glucose intolerance and several other obesity-related chronic complications, affecting virtually all organs and significantly deteriorating the quality of life, which constitute a serious social and economic problem [3].” Page 1, Lines 34-37.

“Men tend towards visceral (android) obesity, that associates with increased insulin resistance and cardio-metabolic risk. Overall, women have higher adiposity than men; however, their adipose tissue accumulates favorably in the subcutaneous depot, which is associated with a lower risk of obesity-related complications [5].” Page 1, Lines 42-45.

“Moreover, recent years have brought evidence confirming the role of estrogen in regulating adipocytes’ insulin sensitivity, metabolism, and secretory activity [8-10].” Page 2, Lines 46-48.